# Scoping Review of the Isometric Mid-Thigh Pull Performance Relationship to Dynamic Sport Performance Assessments

**DOI:** 10.3390/jfmk7040114

**Published:** 2022-12-15

**Authors:** Garrett Giles, Greg Lutton, Joel Martin

**Affiliations:** School of Kinesiology, George Mason University, Manassas, VA 20109, USA

**Keywords:** power, strength, rate of force development

## Abstract

Attempting to understand on-field sport performance from dynamic performance tests of athleticism (i.e., sprinting, jumping, strength) is common practice in sport. In recent years, the isometric mid-thigh pull (IMTP) has gained popularity in the sport performance community as an assessment tool. This scoping review examined the relationship of the IMTP to common dynamic sports assessments to evaluate the robustness of the IMTP to profile lower body force production characteristics. The literature search was conducted according to PRISMA-ScR guidelines. Articles were selected from 5 electronic databases. Data was extracted and synthesized to evaluate the reported relationships between IMTP and common dynamic sport performance assessments. Forty-eight publications were identified and included in the review. Articles reviewed were all within the past 25 years with most (66.7%) published within the past 5 years. Multiple researchers utilized the IMTP across numerous sports and generally reported consistent results. Strong correlations (41.8% of reported, r = 0.71 to 1.00) between the IMTP and the dynamic sport performance assessments were found. The available evidence suggests the IMTP is a viable option for practitioners and researchers to use to profile athletic ability. Furthermore, based on the publication year of included articles, IMTP research is relatively young and warrants further investigation.

## 1. Introduction

Strength and conditioning coaches, sport coaches, and sport scientists often use assessments to evaluate and understand individual performance, monitor recovery and readiness, as well as track progress during training programs. Identifying biomotor abilities that are below minimum thresholds or in general insufficient for success in sport is necessary to ensure training programs focus on the appropriate biomotor abilities. Focusing on biomotor abilities that are already sufficient or focusing on too many biomotor abilities are not the most efficient training program strategies to improve athletic performance [1]. Typical assessment areas of interest include agility/change of direction (COD), strength, speed, power, and endurance assessments [2,3]. Testing results can be either compared to normative data and/or historical data of an athlete then to determine areas of strengths and weaknesses [2,4].

It is common to hold an athlete testing day, and ongoing athlete re-testing over multiple years, where the athletes will perform various performance tests associated with specific athletic attributes believed to be necessary for success in a given sport. It is important to select tests that not only measure what they are intended to measure (i.e., valid), but also be able to measure it consistently (i.e., reliable) [4]. Testing days are not only time consuming and fatiguing, but traditional assessments such as a sprint (maximum speed) or maximum back squat (maximum strength) often only provide one metric per assessment. Another concern may be the safety of maximal dynamic assessments as the injury risk is elevated especially when considering youth or any populations unfamiliar with the nature of maximal dynamic testing [5,6]. The increased risk of injury during a maximal dynamic assessment can occur due to the 1-repetition maximum exceeding tensile strength of structural components [7]. Additionally, during a 1-repition maximum, blood pressure increases past what is normally met using submaximal weights [7]. Moreover, maximum dynamic assessments are considered highly specialized skills that require strong technique to not only protect against injuries, but the attempt accurately reflects the athletes true performance ability [7].

As sports and technology continue to grow, a fast and reliable method to assess athletes’ performance on a regular basis is becoming more feasible. The isometric mid-thigh pull (IMTP) using a force plate is a relatively newer method for determining maximal strength [8,9]. These devices have been reported to be valid and reliable for determining lower extremity maximal strength and peak rate of force development [9,10]. However, methods for calculating rate of force development (RFD) can influence the reliability [11]. Moreover, new research suggests correlation of IMTP metrics (isometric maximum strength and peak rate of force development) to sprint, power, and agility performance [12,13]. An attractive aspect of the IMTP is the ability, due to force plate technology, to provide more than one metric associated with athletic performance, such as maximum force production, time to maximum rate of force development, and rate of force development (RFD) at different time intervals (i.e., 50, 100, 200 ms) [4,12,13].

A practical advantage to using IMTP is attributed to requiring little to no skill to perform the assessment [4,8]. It is common for traditional maximum strength assessments to require previous training and practicing proper technique to ensure safety and optimal performance to acquire valid and reliable data. Several studies demonstrate learning the IMTP can be done in one session thus IMTP test familiarity can be conducted on testing day without negatively impacting the integrity of the assessment [8,9]. Therefore, the IMTP is attractive to coaches from the perspective of being an informative single session assessment that can be performed on a regular basis with little to no skill to perform the assessment, ultimately ensuring safety, saving time, and energy of the athletes. Recent studies explore the nature of IMTP as it relates to dynamic performance [14,15,16].

Over the past decade there has been an increased focus on sport science researchers utilizing the IMTP as an assessment of athlete populations in scientific studies. The scoping review was undertaken to identify and categorize existing research relating the IMTP to dynamic sports assessments and identify gaps in the literature that warrant future research [17,18]. This step is necessary before more focused systematic reviews can be conducted. Therefore, the purpose of this scoping review is twofold. The first aim is to describe the existing body of literature in which the IMTP has been used as an assessment of athletic ability. The second aim is to provide a preliminary report of which traditional dynamic performance assessments are associated with IMTP performance metrics. The findings of the present scoping review can then inform future research, including systematic reviews, meta-analyses and original research.

## 2. Materials and Methods

### 2.1. Protocol and Registration

The scoping review was conducted and reported utilizing the Preferred Reporting Items for Systematic Reviews and Meta-Analysis extension for Scoping Review guidelines [19] (PRISMA-ScR) as well as the guidelines suggested by Arksey and O’Malley [20].

### 2.2. Eligibility Criteria

Eligibility criteria was determined by utilizing the following inclusion criteria: peer reviewed articles published in the past 25 years (1997–2022); IMTP as an assessment tool in recreational and/or competitive athletic population and compared to at least one of the following performance attribute categories: agility/COD, strength, speed, power, or endurance; IMTP metrics such as RFD, and maximum force development in athletics and correlation to performance attributes from one of the following categories: agility, COD, strength, speed, power, or endurance. Exclusion criteria included: non-English language; non-athlete population; unpublished or non-peer reviewed articles; scoping reviews; surgical or rehabilitation interventions and IMTP; other isometric assessments such as grip strength or squat.

### 2.3. Information Sources and Searches

Two researchers (G.G. and G.L.) conducted the literature search to identify, screen, and select the studies to be included in this scoping review. The following multi-step approach was utilized:An initial pilot search was conducted in Google Scholar using the following headings of ‘IMTP’ combined with ‘performance assessments’, ‘performance variables’, ‘performance attributes’, ‘strength’, ‘agility’, ‘COD’, ‘speed’, ‘endurance’, and ‘power’.Identify keywords and search terms that were concise and relevant to IMTP and performance attributes in one of the following categories: agility, COD, strength, speed, power, or endurance.Conduct final search strategy and further backwards search strategies of reference lists from final selection of articles.

The final search strategy was conducted using following databases: SPORTDiscus, CINAHL Plus with Full Text, Human Kinetics Journals, Medline with Full Text, and Google Scholar. When full text was not available from internet sources authors’ (G.G. and G.L.) utilized institutional journal subscriptions to obtain the full-texts. The following search terms were used to perform the final search strategy: relationship between “isometric mid-thigh pull” or “isometric midthigh pull” and performance variables; as well as relationship between “isometric mid-thigh pull” or “isometric midthigh pull” and “strength”, “power”, “agility”, “speed”, and “endurance”. Synonyms of relationship such as association and correlation were used in an attempt to increase search results as well as synonyms for variables such as attributes and assessments; ultimately returning less search results and/or duplicates of final search terms utilized in final search strategy. The search was conducted during March 2022.

### 2.4. Selection of Sources of Evidence

Authors (G.G. and G.L.) evaluated the final search results after removing duplicates by title and abstracts. Full texts were independently reviewed and discussed by authors (G.G. and G.L.) to ensure accuracy and significance as they pertain to our scoping review objectives. In the event of any disagreement, studies would be reviewed by a 3rd author (J.M.) to solve any disagreements. No disagreements were found within the inclusion of articles. 

### 2.5. Data Charting Process

Data charting was performed by authors (G.G. and G.L.) using a custom designed form specifically for extraction of data relevant to objectives of the scoping review (Table 1). Data charting and extraction was performed by dividing the final search results into half and then switching and checking both authors’ charting and extraction. 

Authors (G.G. and G.L.) assessed each study for objective measures (maximum force development, RFD, peak power, dynamic strength, isokinetic strength, jump height, sprint times, agility, and COD times). In cases where study performance assessments, interventions, or correlations were poorly described or required deeper investigations potentially past the individual studies purpose, we attempted to evaluate the results and draw comparisons relevant to the objectives of this scoping review.

### 2.6. Data Items

The following data was extracted and categorized as the following: author, year, study population, outcome variables, and whether significance was found. Definitions of outcome variables are provided in Table 2.

### 2.7. Critical Appraisal of Individual Sources of Evidence

A critical appraisal of individual sources of evidence was not conducted in this scoping review using a standard study quality assessment instrument. The authors did ensure during the full-text review that the procedures used for the IMTP and dynamic performance assessments were consistent with current best practices [4].

### 2.8. Synthesis of Results

A descriptive-analytical method was utilized to synthesize the results based on (1) body of literature retrieved from the scoping review and (2) outcome measures relating to the relationship between IMTP and various performance variables or measures.

## 3. Results

### 3.1. Selecting Sources of Evidence

Initial search results using above defined search parameters returned 387 articles (Figure 1). The authors included thirteen other relevant studies which were found by reverse searching references of the initial search results. Of these 400 articles, 276 duplicates were removed, resulting in 124 articles ready for a first screen pass where the authors reviewed the titles and abstracts to determine if they met the eligibility criteria. There were 53 articles which were fully evaluated for inclusion criteria and 48 met the eligibility criteria to be included in this scoping review.

### 3.2. Characteristics of Sources of Evidence

In all 48 studies, comparisons were made between IMTP to other performance metric types and in some cases, multiple types were compared to assess correlations (Figure 2). Multiple studies reported the relationship between IMTP to the rate of force development of RFD (n = 32) and jump height collectively (n = 33). Regarding jump height, both the counter-movement jump (CMJ) and the squat jump (SJ) were investigated. These studies are of particular interest due to comparison of isometric maximum force (IMTP) to ability to express force quickly (i.e., CMJ and SJ) in movements commonly performed by athletes. The remaining articles investigated relationships to sprinting, COD, peak power, isokinetic strength, and dynamic strength. Of the 48 articles, 17 investigated the IMTP to sport specific movements such as Olympic weightlifting and other sports.

Athletes from a variety of sports were represented in the included studies (Figure 3). A majority of the of the included studies are from within the last 5 years (66.7%), highlighting a recent interest and investigation of the IMTP as an assessment method of to quantify athletic abilities desirable for optimal sport performance (Figure 4).

### 3.3. Results of Individual Sources of Evidence and Synthesis of Results

The results from each included study regarding associations between IMTP and performance assessments are presented in Table 1. Ten areas of interest were found to be consistent in each of the studies (Table 3). Forty-three, or 89.58%, of the included studies were found to report a statistically significant relationship between an IMTP metric and a dynamic athletic performance assessment. Overall, there were 10 variables investigated that included IMTP RFD as it relates to jump height and distance, peak power during CMJ, SJ and dynamic weightlifting, and sport specific assessments. In all cases with more than one reported association between IMTP and a dynamic performance assessment, a majority of the associations were statistically significant (Table 3). This includes a range of assessments associated with desirable athletic abilities of strength, power, vertical and horizontal jumping, sprinting and COD.

## 4. Discussion

### 4.1. Summary of Evidence

This scoping review investigated the body of literature regarding the IMTP performance to traditional dynamic performance assessments. The main finding was that much of the research activity has occurred in the past 5 years and for many specific sports a limited number of studies have been published. However, when considering the overall findings reported in studies there is support for using the IMTP in place of dynamic performance assessments. Without an in-depth analysis of specific findings of studies, the IMTP appears to be a robust assessment of dynamic performance as a variety of dynamic athletic performance assessments were strongly associated with IMTP (Table 3). To this end, a majority of the included studies indicate that the IMTP is an informative assessment methodology [8,10]. Considering that many sports require a variety of biomotor abilities to be tested, this makes the IMTP advantageous due to it being a single assessment associated with numerous dynamic sport assessments. Additionally, the IMTP requires relatively little skill to perform and collect quality data [4,8]. 

### 4.2. Strengths, Limitations and Future Directions

The strengths of the existing literature identified in the scoping review are the number of published articles that met eligibility criteria and the frequency of significant relationships across included studies. While the quality of the studies was not assessed, many of the senior authors were well-established researchers with significant practical experience in the field of sport science. Examples include Stone et al. [13], Comfort et al. [25,26], and Haff et al. [12].

Several limitations must be acknowledged. The existing literature identified in the scoping review indicates the majority of articles were published within the past 5 years and thus this area of inquiry is relatively young in the field of strength and conditioning. Considering the timespan from the start of research to the time findings are published, there is a delay before the findings are applied to sports coaches and strength and conditioning professionals. Research shows the nature of database searching has its limitations with respect to the individual databases being searched [27]. There will likely be more studies investigating the use of the IMTP and its metrics as it relates to dynamic sports assessment in the future. Additionally scoping reviews are meant to investigate existing bodies of literature and identify gaps for future investigation [17,18]. The scoping review methodology utilized provided limited understanding of potential differences in the relationships reported in studies.

Several areas of future investigation for focused systematic reviews can be recommended based on the present scoping review. There needs to be future research on factors such as age, sex, and sport to determine if the relationships reported in the scoping review are truly universal. Additionally, most of the included studies were cross-sectional and only investigated the relationship between IMTP and athletic attributes at one point in time. Recently, literature has reported changes in IMTP variables across periods of training with findings supporting changes in IMTP are related to changes in dynamic sport assessments [26,28,29]. Future studies should continue this areas of inquiry to further assess how changes in IMTP correspond to changes in athletic attributes.

## 5. Conclusions

This scoping review identified a substantial body of literature reporting strong correlations between the IMTP and the dynamic performance metrics. Specifically with regard to peak power, RFD and dynamic strength which are also seen in a recent studies and systematic review [14,15,16]. As technology advances and is more accessible to practitioners, the inclusion of the IMTP is a viable option for profiling athletic abilities. Given the ease with which the IMTP can be administered and association with an array of traditional performance assessments, practitioners should consider adopting the IMTP into assessment batteries. Researchers should expand upon the scoping review by conducting focused systematic reviews and original studies to address further assess the IMTP assessment to profile biomotor abilities.

## Figures and Tables

**Figure 1 jfmk-07-00114-f001:**
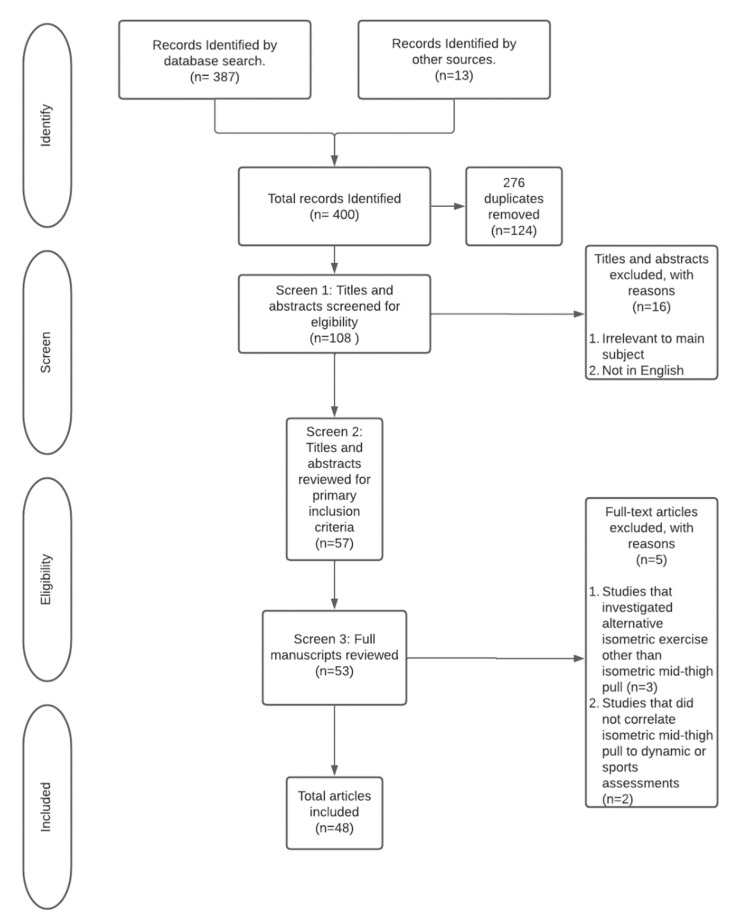
PRISMA flow diagram detailing inclusion and exclusion process.

**Figure 2 jfmk-07-00114-f002:**
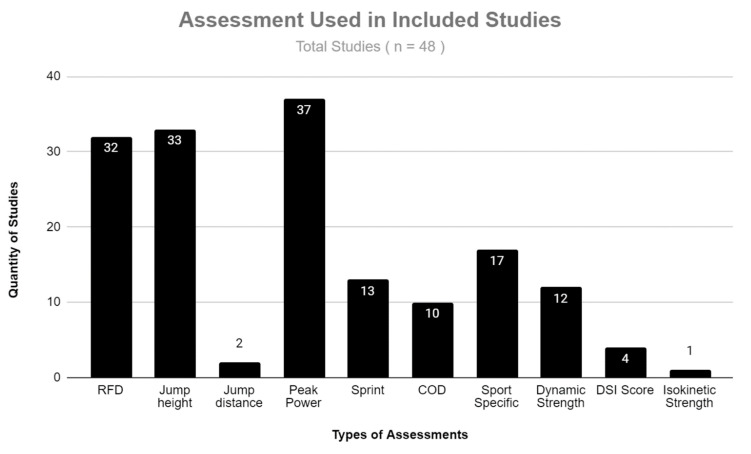
Distribution of types of assessments used in included studies.

**Figure 3 jfmk-07-00114-f003:**
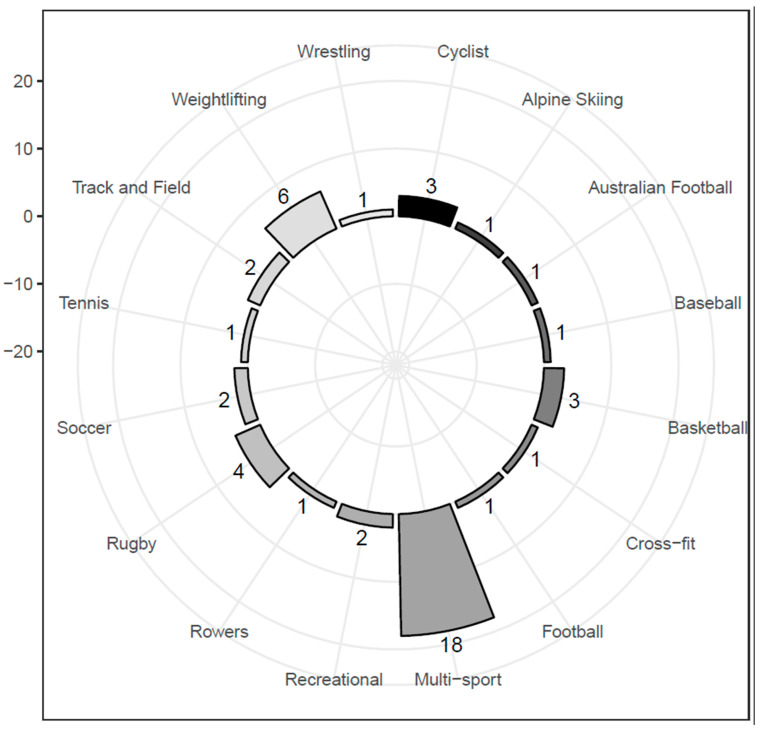
Sport distribution by study.

**Figure 4 jfmk-07-00114-f004:**
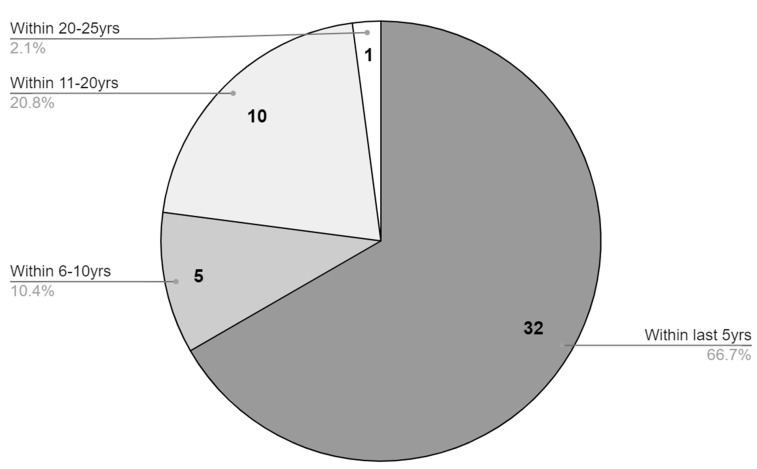
Distribution of studies by year.

**Table 1 jfmk-07-00114-t001:** Summary of significant relationships between isometric mid-thigh pull and dynamic sport assessment performance.

IMTP Comparison	# of Studies	# Statistically. Significant Comparisons	Significance %	# Weak Associations	# Moderate Associations	# Strong Associations
RFD	32	31	96.88%	2	16	13
Jump height	33	30	90.91%	2	17	11
Jump distance	2	2	100.00%	0	1	1
Peak Power	22	22	100.00%	2	17	3
Sprint	13	12	92.31%	1	8	3
COD	10	9	90.00%	0	7	2
Sport Specific	17	16	94.12%	0	10	6
Dynamic Strength	12	12	100.00%	1	6	5
DSI Score	4	3	75.00%	0	1	2
Isokinetic Strength	1	1	100.00%	0	0	1
Total Studies	48	43	89.58%	1	24	18

**Table 2 jfmk-07-00114-t002:** Definitions of categories of outcome variables used to characterize distribution of studies [21,22,23].

Terminology	Definition
Isometric Strength	Maximal force produced during muscular contraction against resistance, without shortening of muscle fibers
Rate of Force Development	Measure of explosive strength or how fast an athlete can develop force
Jump Height	Vertical displacement from take-off to vertex. Can include displacement and reach depending on testing method
Jump Distance	Horizontal displacement from take-off to landing
Peak power/force	Maximum force of torque developed during muscle action over a given amount of time
Sprint	Maximum speed over specific distances. Measured as time to completion
Change of Direction	Ability to change movement direction, velocities, or modes. Measured as time to completion
Dynamic Strength	Isotonic strength movements. Typically, measured as a repetition maximum
Dynamic Strength Index	Difference between minimum and explosive strength capacity. Usually, quantified as the ratio between IMTP_peak force_ and CMJ_peak force_
Sport Specific Task	Performance movements that specifically mimic a task or skill in sport
Isokinetic Strength	Strength training in which speed of movement remains constant, but resistance varies

**Table 3 jfmk-07-00114-t003:** Analysis of significance of areas of interest. Abbreviations: RFD, rate of force development; COD, change of direction; DSI, dynamic strength index. Strength of associations were determined as: weak—r = 0.10–0.40 denoted by “●”; moderate—r = 0.41–0.70 as denoted by “‡”; strong—r ≥ 0.71 as denoted by “☆.” Insignificant or lack of associations are denoted as ”x.” [24].

Author	RFD	Jump height	Jump distance	Peak Power	Sprint	COD	Sport Specific	Dynamic Strength	DSI Score	Isokinetic Strength	Signifcance?
Bailey (2013)		‡		‡							‡
Beattie (2017)		‡		‡			‡				‡
Beckham (2013)	☆						☆				☆
Beckham (2019)	☆	☆									☆
Bourgeois (2017)					x	x	x				
Brady (2020)	‡				‡		‡				‡
Comfort (2020)	☆										☆
Comfort (2018)		☆						☆	☆		☆
Cross (2021)	‡	‡					‡				‡
Cunningham (2018)		‡					‡				‡
Dobbs (2020)		‡		‡				‡			‡
Dos’Santos (2017)	☆	☆		☆							☆
Guppy (2018)	‡			‡							‡
Haff (1997)	☆			☆			☆				☆
Haff et al. (2005)	☆			☆			☆	☆			☆
Hayes (2018)	☆	☆					☆				☆
Hornikova (2021)	‡	‡		‡	‡	‡					‡
Hornsby (2017)	‡	‡		‡			‡				‡
Hornsby (2021)							☆				☆
Kawamori (2006)	☆	☆		☆			☆	☆			☆
Khamoui (2011)	‡	‡		‡							‡
Kozinc (2021)	‡			‡		‡	‡				‡
Kraska et al. (2009)	‡	‡		‡							‡
Kuki et al. (2017)	☆	☆		☆	☆						☆
Lee and Kim (2020)		☆								☆	☆
Mangine (2020)				x							
Mason (2021)	‡	‡		‡	‡	‡					‡
McGuigan (2008)	☆	☆						☆			☆
McGuigan (2006)	☆							☆			☆
McMahon (2017)		☆							☆		☆
Merrigan (2020)	x	x									
Norris (2021)	‡	‡									‡
Nuzzo (2008)	●	●		●				●			●
Pichardo (2019)		‡	‡		‡						‡
Post et al. (2022)		‡	‡	‡	‡	‡	‡				‡
Spiteri et al. (2014)				☆		☆					☆
Stone et al. (2004)	☆			☆			☆	☆			☆
Stone (2003)	☆	☆		☆	☆		☆				☆
Suchomel (2016)	‡	‡			‡	‡	‡	‡			‡
Suchomel (2020)	‡	‡		‡					‡		‡
Thomas (2018)		x				x					
Thomas (2017)		x							x		
Thomas (2017)		‡									‡
Townsend (2019)	‡	‡			‡	‡		‡			‡
Travis (2018)	‡	‡						‡			‡
Vercoe (2018)	☆				☆						☆
Wang (2016)	☆				☆	☆		☆			☆
West et al. (2011)	☆	☆		☆	☆						☆

## Data Availability

Not applicable.

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
