# Peer review of "Scoping Review of the Isometric Mid-Thigh Pull Performance Relationship to Dynamic Sport Performance Assessments"

_jfmk, 2022, doi:10.3390/jfmk7040114_

Round 1

Reviewer 1 Report

I commend the authors on the topic given the growing interest of the IMTP and the increased accessibility of technology to conduct the assessment. I'm just not sure there is enough detail in this review to help practitioners understand the value (or limitations) of the IMTP. If one of the purposes is to describe the existing body of literature, more work needs to be done (age, gender, level of athlete, variables from the IMTP used in these studies). In addition, there is quite a bit of grammatical work to be done in terms of punctuation, flow, and the removal of superfluous content.

Abstract:

Line 18: Please provide specific correlation values (or the range) between IMTP and the dynamic assessments

Line 21: Please include a comma after ‘articles’

Introduction:

Line 28: Please remove comma after ‘sport’

Line 34-36: Please remove the second ‘then’ as it is not needed.

Line 41-43: The number of metrics are more a result of the technology being used to assess a quality, as opposed to the assessment picked. Additionally, testing days may not always be time consuming or fatiguing, depending on the tests being performed. Please revise this sentence for clarity.

Line 43: Is there a reference to support that the injury risk is elevated during a dynamic assessment?

Line 51: Once again, the ability to provide more metrics is because it is using a dynamometer, not the IMTP itself.

Line 51-55: I would combine these two thoughts into one sentence for improved flow: …one metric associated with athletic performance, such as maximum force production, time to maximum…

Line 67-70: There is little information in the introduction leading to these purposes, as it mostly pertains to testing in general and why one would choose the IMTP. Please provide some relevant information as to why describing the existing body of literature is important or why reporting which dynamic tests are related to IMTP performance is important.

Line 84: Is this meant to say ‘non-peer-reviewed articles’?

Line 103-104: Based on these search terms, we might’ve missed articles that used ‘isometric midthigh pull’ in the title - same test, just no hyphen (West, Daniel J., et al. "Relationships between force–time characteristics of the isometric midthigh pull and dynamic performance in professional rugby league players." The Journal of Strength & Conditioning Research 25.11 (2011): 3070-3075). Please re-conduct search and include any other relevant articles.

Line 161: Please add a comma after the second ‘height’ in this line.

Line 165: The ‘COD’ abbreviation has already been defined and is not needed here.

Line 171-173: Please add a comma after ‘(76.3%)’ and rephrase end of sentence. Just because there are correlations with dynamic sport performance, the IMTP is still an isometric assessment that may or may not be related to dynamic actions.

Line 186-187: Please spell out numbers under ten (1 to one) and add a comma after ‘assessment’.

Line 196-199: The review did not investigate relationships, it merely stated if other studies found relationships. The review also did not discern which IMTP metrics (absolute and/or relative peak force, RFD, impulse) are related to the (very broad) measures of dynamic performance. Line 198 says ‘The majority of the included studies utilize the IMTP and deem the use of the IMTP metric as an effective assessment methodology’ - first, ALL the studies utilized the IMTP, right? Second, the IMTP itself is not a metric, it is an assessment that provides metrics.

Line 201: Reference to 10, 16 at the end of the sentence do not match the rest of the text.

Line 205: Many of the sports only included one study, so not sure that qualifies as being robust. This line also refers to the backgrounds of the included studies, which we have no info about. Is it more or less accurate in males vs females? Youth vs adults? Is RFD more strongly correlated than peak force to the dynamic performance metrics?

Line 221: Please add a comma after ‘assessed’ and use the citations in the following sentence at the end of this one - no need to include a sentence with examples.

Author Response

Reviewer #1:

I commend the authors on the topic given the growing interest of the IMTP and the increased accessibility of technology to conduct the assessment. I'm just not sure there is enough detail in this review to help practitioners understand the value (or limitations) of the IMTP. If one of the purposes is to describe the existing body of literature, more work needs to be done (age, gender, level of athlete, variables from the IMTP used in these studies). In addition, there is quite a bit of grammatical work to be done in terms of punctuation, flow, and the removal of superfluous content.

Response: Thank you for your time reviewing the manuscript and feedback provided. The comments below are appreciated. We believe that the feedback has helped to improve the paper. One aspect that we would like to clarify is our decision to conduct a scoping review as opposed to a systematic review with a meta-analysis. In recent years there has been an increase in publications/interest in the IMTP as we’ve observed based on published literature. Particularly, in regards to using the IMTP as an assessment tool for sports performance. To that end our inquiry was focused on characterizing the research activity by reporting on the body of literature. This will allow more focused systematic reviews and meta-analyses can be conducted. Our decision to conduct a scoping review was based on a paper by Arksey and O’Malley which provided 4 reasons to undertake a scoping review as well as differentiated a scoping review from a systematic review.

Arksey, H.; O’Malley, L. Scoping Studies: Towards a Methodological Framework. International Journal of Social Research Methodology 2005, 8, 19–32, doi:10.1080/1364557032000119616.

Ultimately, our scoping review was conducted to inform the research community as opposed to practitioners. As this is a scoping review, data extraction is outside the scope of a scoping review and would be more suited to a systematic or meta-analysis. Our intent was more so to describe the research activity and high-level findings than to scrutinize specific results reported in the existing literature. We believe this was not clearly articulated in our original manuscript and have made edits to clarify.

Abstract:

Line 18: Please provide specific correlation values (or the range) between IMTP and the dynamic assessments

Response: Thank you for the suggestion. The values have been added.

Line 21: Please include a comma after ‘articles’

Response: Fixed

Introduction:

Line 28: Please remove comma after ‘sport’

Response: Fixed.

Line 34-36: Please remove the second ‘then’ as it is not needed.

Response: Fixed.

Line 41-43: The number of metrics are more a result of the technology being used to assess a quality, as opposed to the assessment picked. Additionally, testing days may not always be time consuming or fatiguing, depending on the tests being performed. Please revise this sentence for clarity.

Response: References were added citing testing days of this nature are time consuming and labor intensive.

Line 43: Is there a reference to support that the injury risk is elevated during a dynamic assessment?

Response: References were added to address this point and text amended to reflect the intent of the statement.

Line 51: Once again, the ability to provide more metrics is because it is using a dynamometer, not the IMTP itself.

Response: The reviewer is correct and this has been clarified.

Line 51-55: I would combine these two thoughts into one sentence for improved flow: …one metric associated with athletic performance, such as maximum force production, time to maximum…

Response: Thank you for the suggestion. This has been corrected.

Line 67-70: There is little information in the introduction leading to these purposes, as it mostly pertains to testing in general and why one would choose the IMTP. Please provide some relevant information as to why describing the existing body of literature is important or why reporting which dynamic tests are related to IMTP performance is important.

Response: References added to reflect current literature addressing the topic.

Line 84: Is this meant to say ‘non-peer-reviewed articles’?

Response: Thank you for the correction. Yes it should say non-peer reviewed articles and is now fixed.

Line 103-104: Based on these search terms, we might’ve missed articles that used ‘isometric midthigh pull’ in the title - same test, just no hyphen (West, Daniel J., et al. "Relationships between force–time characteristics of the isometric midthigh pull and dynamic performance in professional rugby league players." The Journal of Strength & Conditioning Research 25.11 (2011): 3070-3075). Please re-conduct search and include any other relevant articles.

Response: The search was reconducted with no hyphen and the studies included was updated to include any additional studies found.

Line 161: Please add a comma after the second ‘height’ in this line.

Response: Fixed.

Line 165: The ‘COD’ abbreviation has already been defined and is not needed here.

Response: Fixed.

Line 171-173: Please add a comma after ‘(76.3%)’ and rephrase end of sentence. Just because there are correlations with dynamic sport performance, the IMTP is still an isometric assessment that may or may not be related to dynamic actions.

Response: Thank you for the feedback. The comma has been added and sentence rephrased.

Line 186-187: Please spell out numbers under ten (1 to one) and add a comma after ‘assessment’.

Response: The edits have been made. Thank you.

Line 196-199: The review did not investigate relationships, it merely stated if other studies found relationships. The review also did not discern which IMTP metrics (absolute and/or relative peak force, RFD, impulse) are related to the (very broad) measures of dynamic performance. Line 198 says ‘The majority of the included studies utilize the IMTP and deem the use of the IMTP metric as an effective assessment methodology’ - first, ALL the studies utilized the IMTP, right? Second, the IMTP itself is not a metric, it is an assessment that provides metrics.

Response: The reviewer is correct and the wording of these statements were incorrect based on our purposes. We’ve edited this section of text to more correctly articulate what was done and reported.

Line 201: Reference to 10, 16 at the end of the sentence do not match the rest of the text.

Response: Thank you for finding this. We have fixed the issue.

Line 205: Many of the sports only included one study, so not sure that qualifies as being robust. This line also refers to the backgrounds of the included studies, which we have no info about. Is it more or less accurate in males vs females? Youth vs adults? Is RFD more strongly correlated than peak force to the dynamic performance metrics?

Response: We’ve made substantial edits to this section of the discussion. Upon review too much was stated when considering our purposes and reporting of only higher level findings. This section is now more concise

Line 221: Please add a comma after ‘assessed’ and use the citations in the following sentence at the end of this one - no need to include a sentence with examples.

Response: Fixed

Reviewer 2 Report

General Comments

This is an interesting and potentially informative scoping review; however, clear rationales need to be provided for certain decisions that have been made, or these decisions amended (please see specific comments below). In addition, methodological considerations when performing the IMTP need to be explored, including the appropriate posture and analysis of the force-time data when considering time related variables, as these have been shown to affect the reliability of the variables and their magnitudes, which will in turn affect the correlations reported. Some discussion as to why some researchers have reported strong relationships between specific IMTP variables and other assessment methods and other have reported no relationship, or weak associations should be considered in the context of the methods that the authors have used. Please also see some specific comments and suggestions below.

Specific Comments

Line 8: These tests are commonly used to determine force production characteristics and how they relate to performance in athletic tasks. They are not / should not be used to attempt to ‘predict’ on-field performance; no appropriate prediction equations for field-based performance from the IMTP have been established. Please amend and reword accordingly here and throughout the manuscript.

Line 12: The IMTP is not used to profile athletic abilities, but to evaluate the force production characteristics of the lower body. Please amend accordingly.

Line 16: Why date restrict to within the last 20 years, when the original publication was published in 1997? This restriction makes no sense, and all manuscripts should be included from 1997 onwards.

Line 17: Multiple sports did not investigate anything. Change to ‘Multiple researchers utilized the IMTP across numerous sports…’

Lines 29-32: This statement should be explained. Why would this occur? Why would any S&C coach focus purely on enhancing maximum power of force-velocity imbalances?

Lines 37-44: This is not the only approach. In some cases ongoing assessments are used and should be acknowledged. While some individuals may be concerned for the safety of maximal dynamic assessments, is there any evidence to suggest that they result in injuries which exceed the injury rates commonly reported from training or competition?

Lines 48-50: Please clarify if it is data from the ‘dynamometers’ / strain gauges, or force plates that demonstrate reliable RFD values, as many strain gauges do not sample at a high enough frequency to obtain reliable and precise measures of RFD.

Lines 51-51: Which IMTP metrics relate to these athletic tasks?

Lines 53-55: You should acknowledge the fact that the methods of calculating RFD may affect the reliability and correlations, see Haff et al. (2015) JSCR. 29(2): 386-395.

Line 57: Please change ‘movement’ to assessment, as the task is isometric and therefore movement should be negligible. Please also amend this on line 64.

Line 78: As mentioned above why date restrict to within the last 20 years, when the original publication was published in 1997? This restriction makes no sense, and all manuscripts should be included from 1997 onwards.

Line 101: Restricting some of these searches to only those with full text available via the data base may have resulted in numerous studies being omitted as some are only available as full text on data bases 12-24 months after publication. What was the rationale for this? Why not search for all studies and access those not available in full text via the authors’ institutional journal subscriptions, institutional repositories, and  and via contacting authors directly?

Table 2: Jump height is jump and reach distance if using a vertec. In most cases sprint will be time to complete a specified distance and not speed and COD will usually be time to completion. DSI is usually the ratio between IMTP peak force and CMJ peak force. Please check each of these and amend accordingly.

Figure 1: Next to Screen 3, there should be a horizontal arrow with another box identifying why some studies were excluded and the number excluded for each reason, in line with PRISMA guidelines.

Line 185: RFD during what task / activity(ies)?

Peak power during what task / activity(ies)?

Lines 186-188: In many of these studies multiple correlations were performed without correcting the resulting p values for familywise error rates. This should be acknowledged, and the p values amended accordingly. In addition, it is the magnitude of the correlation, based on the r value, that is most important and should be the focus, along with the associated coefficient of determination where Pearson’s correlations were performed.

Lines 234-236: Numerous researchers have published manuscripts detailing the changes in IMTP variables across periods of training, which should be acknowledged here, for example a quick search on PubMed just revealed numerous studies including Comfort et al. (2018) JSCR. 32 (8): 2116-2129, Suchomel et al. (2020) JSCR. 34(7): 1808-1818; Comfort et al. (2022) JSCR. 36(3): 593-599 to name a few.

Author Response

Reviewer #2

This is an interesting and potentially informative scoping review; however, clear rationales need to be provided for certain decisions that have been made, or these decisions amended (please see specific comments below). In addition, methodological considerations when performing the IMTP need to be explored, including the appropriate posture and analysis of the force-time data when considering time related variables, as these have been shown to affect the reliability of the variables and their magnitudes, which will in turn affect the correlations reported. Some discussion as to why some researchers have reported strong relationships between specific IMTP variables and other assessment methods and other have reported no relationship, or weak associations should be considered in the context of the methods that the authors have used. Please also see some specific comments and suggestions below.

Response: Thank you for your review and feedback on the scoping review. Particularly the note about the rationale for the date of the search was valuable. We have reconducted the search and updated our results since the first review of the paper. There have been other substantial edits throughout the manuscript to address the comments of both reviewers. One aspect that we would like to clarify is our decision to conduct a scoping review as opposed to a systematic review with a meta-analysis. In recent years there has been an increase in publications/interest in the IMTP as we’ve observed based on published literature. Particularly, in regards to using the IMTP as an assessment tool for sports performance. To that end our inquiry was focused on characterizing the research activity by reporting on the body of literature. This will allow more focused systematic reviews and meta-analyses can be conducted. Our decision to conduct a scoping review was based on a paper by Arksey and O’Malley which provided 4 reasons to undertake a scoping review as well as differentiated a scoping review from a systematic review.

Arksey, H.; O’Malley, L. Scoping Studies: Towards a Methodological Framework. International Journal of Social Research Methodology 2005, 8, 19–32, doi:10.1080/1364557032000119616.

Ultimately, our scoping review was conducted to inform the research community as opposed to practitioners. As this is a scoping review, data extraction is outside the scope of a scoping review and would be more suited to a systematic or meta-analysis. Our intent was more so to describe the research activity and high-level findings than to scrutinize specific results reported in the existing literature. We believe this was not clearly articulated in our original manuscript and have made edits to clarify.

To this end some of the specific edits that the reviewer provides, we agree are important to address, but would be better suited for future systematic reviews and original research studies.

Specific Comments

Line 8: These tests are commonly used to determine force production characteristics and how they relate to performance in athletic tasks. They are not / should not be used to attempt to ‘predict’ on-field performance; no appropriate prediction equations for field-based performance from the IMTP have been established. Please amend and reword accordingly here and throughout the manuscript.

Response: We have reworded to make sure that it is clear we are not attempting to ‘predict’ on-field performance throughout the manuscript.

Line 12: The IMTP is not used to profile athletic abilities, but to evaluate the force production characteristics of the lower body. Please amend accordingly.

Response: Fixed

Line 16: Why date restrict to within the last 20 years, when the original publication was published in 1997? This restriction makes no sense, and all manuscripts should be included from 1997 onwards.

Response: Search was expanded to include earlier studies

Line 17: Multiple sports did not investigate anything. Change to ‘Multiple researchers utilized the IMTP across numerous sports…’

Response: Thank you fixed.

Lines 29-32: This statement should be explained. Why would this occur? Why would any S&C coach focus purely on enhancing maximum power of force-velocity imbalances?

Response: This statement has been reworded to improve the intended meaning.

Lines 37-44: This is not the only approach. In some cases ongoing assessments are used and should be acknowledged. While some individuals may be concerned for the safety of maximal dynamic assessments, is there any evidence to suggest that they result in injuries which exceed the injury rates commonly reported from training or competition?

Response: Thank you for the response. Citations and additional comments have been added

Lines 48-50: Please clarify if it is data from the ‘dynamometers’ / strain gauges, or force plates that demonstrate reliable RFD values, as many strain gauges do not sample at a high enough frequency to obtain reliable and precise measures of RFD.

Response: The comment is well-received and it was intended to say force plates. The reviewer makes an important point and we appreciate brining attention to this matter.

Lines 51-51: Which IMTP metrics relate to these athletic tasks?

Response: Fixed

Lines 53-55: You should acknowledge the fact that the methods of calculating RFD may affect the reliability and correlations, see Haff et al. (2015) JSCR. 29(2): 386-395.

Response: Thank you for the feedback. We have added a sentence to acknowledge the methods of calculating the RFD can affect the reliability.

Line 57: Please change ‘movement’ to assessment, as the task is isometric and therefore movement should be negligible. Please also amend this on line 64.

Response: Edit has been made.

Line 78: As mentioned above why date restrict to within the last 20 years, when the original publication was published in 1997? This restriction makes no sense, and all manuscripts should be included from 1997 onwards.

Response: The reviewer provided valuable feedback. As mentioned above we changed the date restriction and re-conducted the search.

Line 101: Restricting some of these searches to only those with full text available via the data base may have resulted in numerous studies being omitted as some are only available as full text on data bases 12-24 months after publication. What was the rationale for this? Why not search for all studies and access those not available in full text via the authors’ institutional journal subscriptions, institutional repositories, and  and via contacting authors directly?

Response: When full text was not available the researchers did utilize institutional journal subscriptions. It was just not previously stated.

Table 2: Jump height is jump and reach distance if using a vertec. In most cases sprint will be time to complete a specified distance and not speed and COD will usually be time to completion. DSI is usually the ratio between IMTP peak force and CMJ peak force. Please check each of these and amend accordingly.

Response: Fixed

Figure 1: Next to Screen 3, there should be a horizontal arrow with another box identifying why some studies were excluded and the number excluded for each reason, in line with PRISMA guidelines.

Response: This has been added.

Line 185: RFD during what task / activity(ies)?

Response: It was IMTP RFD. This has been added.

Peak power during what task / activity(ies)?

Response: The activities were countermovement jump, static jump and dynamic weightlifting. This has been added.

Lines 186-188: In many of these studies multiple correlations were performed without correcting the resulting p values for familywise error rates. This should be acknowledged, and the p values amended accordingly. In addition, it is the magnitude of the correlation, based on the r value, that is most important and should be the focus, along with the associated coefficient of determination where Pearson’s correlations were performed.

Response: Thank you for this comment. We took this comment into consideration and in the spirit of conducting a scoping review vs. a systematic review we felt that providing too much detail regarding the actual results from studies would detract from our intent of describing the research activity relating IMTP to dynamic sport performance assessments. What the reviewer suggests would be better suited for a focused systematic review in the future.

Lines 234-236: Numerous researchers have published manuscripts detailing the changes in IMTP variables across periods of training, which should be acknowledged here, for example a quick search on PubMed just revealed numerous studies including Comfort et al. (2018) JSCR. 32 (8): 2116-2129, Suchomel et al. (2020) JSCR. 34(7): 1808-1818; Comfort et al. (2022) JSCR. 36(3): 593-599 to name a few.

Response: Thank you for this feedback. The statement has been revised and citations added.

Round 2

Reviewer 1 Report

Thank you for the clarification and changes made - I believe the quality of the manuscript has been improved.

Author Response

Thank you again for your feedback and time reviewing our paper.

Reviewer 2 Report

The authors should be commended on the thorough revisions to the manuscript.

Author Response

(The authors gave the same response as above.)
